

# How beavers affect riverine aquatic macroinvertebrates: a review

Susan Washko[1], Nigel Willby[2] and Alan Law[2]

[1] School of Natural Resources and the Environment, University of Arizona, Tucson, Arizona, United States
[2] Biological and Environmental Sciences, University of Stirling, Stirling, Scotland, United Kingdom

## ABSTRACT

**Background:** As ecosystem engineers, the construction of dams by beavers alters stream habitat physically and biologically, making them a species of interest for habitat restoration. Beaver-created habitat changes affect a wide range of aquatic invertebrate species. However, despite numerous individual studies of how beavers affect aquatic macroinvertebrate assemblages, there has been no evaluation of the consensus of these effects across studies.

**Methodology:** We collated and examined studies comparing beaver-created ponds to nearby lotic reaches to determine general trends in aquatic macroinvertebrate richness, density, biomass, and functional composition between habitats. From this evidence, we highlight knowledge gaps in how beaver activity affects aquatic macroinvertebrates.

**Results:** Overall, in the majority of studies, aquatic macroinvertebrate richness was higher in nearby lotic reaches compared to beaver-created ponds, but richness at coarser scales (gamma diversity) increased with the addition of beaver ponds due to increased habitat heterogeneity. Functional feeding group (FFG) patterns were highly context-dependent, though predator taxa were generally more abundant in beaver ponds than adjacent lotic reaches. Site-specific geomorphological changes, coupled with dam or riparian zone characteristics and resulting differences in basal food resources likely shape other FFG responses.

**Conclusions:** We identify a lack of long-term studies at single or multiple sites and conclude that fine-scale approaches may improve our understanding of the dynamics of macroinvertebrates within the freshwater realm and beyond. Due to the context-dependent nature of each study, further systematic studies of beaver engineering effects across a wider variety of environmental conditions and wetland types will also help inform land and species management decisions, such as where to prioritize protection of beaver habitats in the face of a global freshwater biodiversity crisis, or where to restore beaver populations to deliver maximum benefit.

Corresponding author
Susan Washko,
swashko@email.arizona.edu

## INTRODUCTION

Once prized more as a commodity than as key components of an ecosystem, beavers were virtually extirpated from North America (*Castor canadensis*) and Eurasia (*Castor fiber*) by

the early 1900s (*Baker & Hill, 2003*; *Halley, Rosell & Saveljev, 2012*). Due to numerous reintroductions, translocations, and legal protection, beavers are now recolonizing these regions, once again coppicing, felling trees and building dams and lodges. Land managers, practitioners and scientists are becoming increasingly interested in how beaver engineering activities alter stream ecosystems within a modern landscape context (*Brazier et al., 2021*). Construction of woody debris dams along small streams restores lost, natural heterogeneity that can improve fish habitat (*Kukuła & Bylak, 2010*; *Malison et al., 2014*; *Bouwes et al., 2016*; *Bylak & Kukuła, 2018*), reduce incision and sedimentation (*Pollock et al., 2014*), assist with flood and drought alleviation (*Law, Mclean & Willby, 2016*; *Puttock et al., 2021*), and increase filtration of nutrients and metals (*Čiuldienė et al., 2020*; *Smith et al., 2020*; *Murray, Neilson & Brahney, 2021*). Therefore, beavers are often considered as agents for stream restoration (*Law et al., 2017*; *Brazier et al., 2021*) with their services being increasingly sought across their former range (*Pollock et al., 2014*; *Bailey, Dittbrenner & Yocom, 2019*).

Construction of dams by beavers and the subsequent habitat changes can have a major effect on biodiversity. Beaver engineering alters habitats through changes to depth, water velocity, benthic substrate composition (*i.e.* interstitial spaces and surfaces for biofilm growth), organic matter availability (*Washko, Roper & Atwood, 2020*), and aquatic plant growth (*Law et al., 2017*). The successional gradient of these physical and biological changes in beaver-altered ecosystems provides a complex mosaic of habitat types that can support numerous aquatic invertebrate taxa (*Bush et al., 2019*; *Bylak et al., 2020*; *Nummi et al., 2021*). Aquatic macroinvertebrates are one of the most predominant groups studied globally in relation to beaver-induced habitat alteration (Table 1). This is because aquatic macroinvertebrates are widely-used indicators of water quality (*Hodkinson & Jackson, 2005*), straightforward to sample, highly diverse, essential to ecosystem functioning (*Wallace & Webster, 1996*), play a major role in the linkages between aquatic, riparian and terrestrial habitats (*Anderson & Rosemond, 2010*) and are a significant food source for various vertebrate consumers (*Nummi, 1992*; *Nummi et al., 2011*; *Kemp et al., 2012*; *McCaffery & Eby, 2016*).

Despite numerous studies detailing beavers' effects on riverine aquatic macroinvertebrates, a synthesis of trends and common findings is lacking, therefore we have a limited knowledge of the transferability of their stream restoration potential. This evidence is crucial to inform further reintroductions or translocations and for protection of beavers and their habitats in the face of a global freshwater biodiversity crisis. The aim of this literature review is to assess patterns of aquatic macroinvertebrate community composition between beaver-created ponds and associated free-running stream segments.

Our research objectives were as follows:

1. Determine if the current literature reveals any generalizable differences in aquatic macroinvertebrate i) taxa richness, ii) density, iii) biomass, and iv) functional feeding groups between beaver ponds and nearby lotic stream segments and describe these differences.

2. Expose research gaps in the beaver pond aquatic macroinvertebrate literature.

**Table 1 Studies included in this review.** Studies comparing the aquatic macroinvertebrate community between beaver ponds and nearby lotic reaches.

|     | Study                              | Location          |
| --- | ---------------------------------- | ----------------- |
| 1   | *Anderson & Rosemond, 2007*        | Chile             |
| 2   | *Arndt & Domdei, 2011*             | Germany           |
| 3   | *Bush et al., 2019*                | Georgia, USA      |
| 4   | *Clifford, Wiley & Casey, 1993*    | Alberta, Canada   |
| 5   | *Gard, 1961*                       | California, USA   |
| 6   | *Harthun, 1999*                    | Germany           |
| 7   | *Hodkinson, 1975b*                 | Alberta, Canada   |
| 8   | *Huey & Wolfrum, 1956*             | New Mexico, USA   |
| 9   | *Kukuła et al., 2008*              | Poland            |
| 19  | *Law, Mclean & Willby, 2016*       | Scotland          |
| 11  | *Malison et al., 2014*             | Alaska, USA       |
| 12  | *Malison & Halley, 2020*           | Norway            |
| 13  | *Margolis, Raesly & Shumway, 2001* | Pennsylvania, USA |
| 14  | *McDowell & Naiman, 1986*          | Quebec, Canada    |
| 15  | *Naiman, McDowell & Farr, 1984*    | Quebec, Canada    |
| 16  | *Pliūraitė & Kesminas, 2012*       | Lithuania         |
| 17  | *Robinson et al., 2020*            | Switzerland       |
| 18  | *Rolauffs, Hering & Lohse, 2001*   | Germany           |
| 19  | *Rupp, 1955*                       | Maine, USA        |
| 20  | *Sprules, 1941*                    | Ontario, Canada   |
| 21  | *Strzelec, Białek & Spyra, 2018*   | Poland            |
| 22  | *Washko, Roper & Atwood, 2020*     | Utah, USA         |
| 23  | *Wojton & Kukuła, 2021*            | Poland            |

# SURVEY METHODOLOGY

The reviewed studies were found through Google Scholar, Scopus, and Web of Science searches for keywords: *beaver pond, beaver dam, macroinvertebrate, aquatic macroinvertebrate, aquatic invertebrate*, and combinations thereof. Papers were selected if they sampled aquatic macroinvertebrates in both an in-stream lentic beaver-created habitat and a nearby, disparate lotic reach where flow was not affected by beaver dams. For example, we included studies where beaver ponds formed behind dams within stream channels, and both the pond and adjacent non-dammed reaches were sampled. Studies of wetlands (*e.g.* research comparing stages of beaver wetland succession, or comparing beaver and non-beaver wetlands) or lakes were omitted, despite these studies also documenting shifts in the biological community following beaver engineering (*Hood & Larson, 2014*; *Bush & Wissinger, 2016*; *Willby et al., 2018*; *Law et al., 2019*; *Bashinskiy, 2020*; *Nummi et al., 2021*). In total, 23 studies from across the globe met our criteria (Fig. 1; Table 1; Table S1). The studies span publication dates from 1941–2021, the most common publication date being the year 2020, indicating this is a growing field of study.

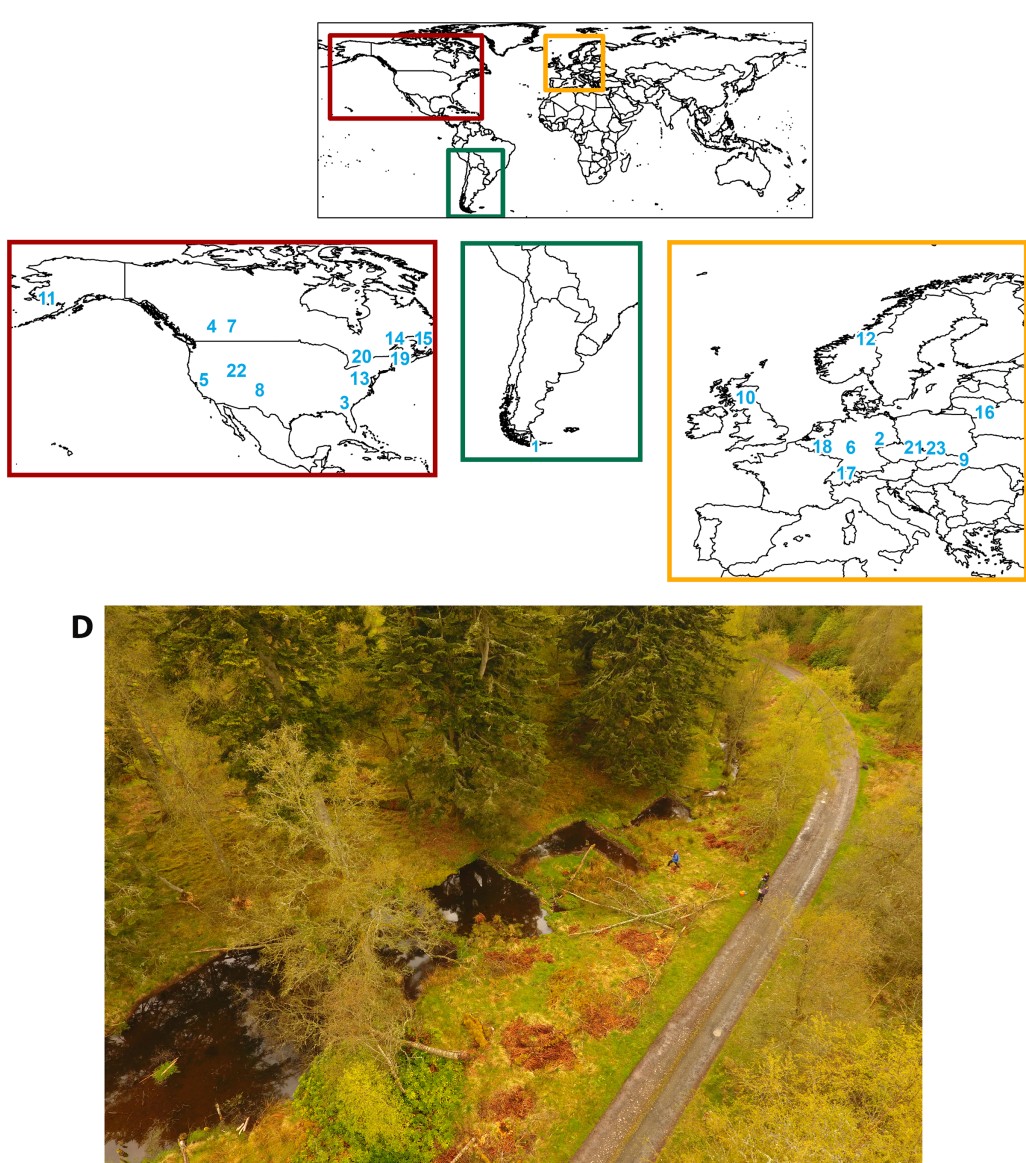

**Figure 1 Study locations included in the review and an example of a beaver-altered stream.** Map of the location for each study reviewed. Study locations spanned three continents: (A) North America (*Castor canadensis*), (B) South America (*Castor canadensis*, nonnative), and (C) Europe (*Castor fiber*). The numbered locations correspond to each study's site, as listed in Table 1. (D) Studies included were for streams containing beaver ponds, such as the ponds in Scotland from study #10, pictured here.

Invertebrate data from these 23 studies were collated and compared to interpret general trends. We used categories such as species richness (number of species occurring in one habitat), density (number of individual macroinvertebrates per square meter of habitat), and biomass (macroinvertebrate mass per square meter of habitat) to compare macroinvertebrate patterns between lotic reaches and beaver ponds. We also recorded whether the authors noted a higher or lower abundance of each functional feeding group

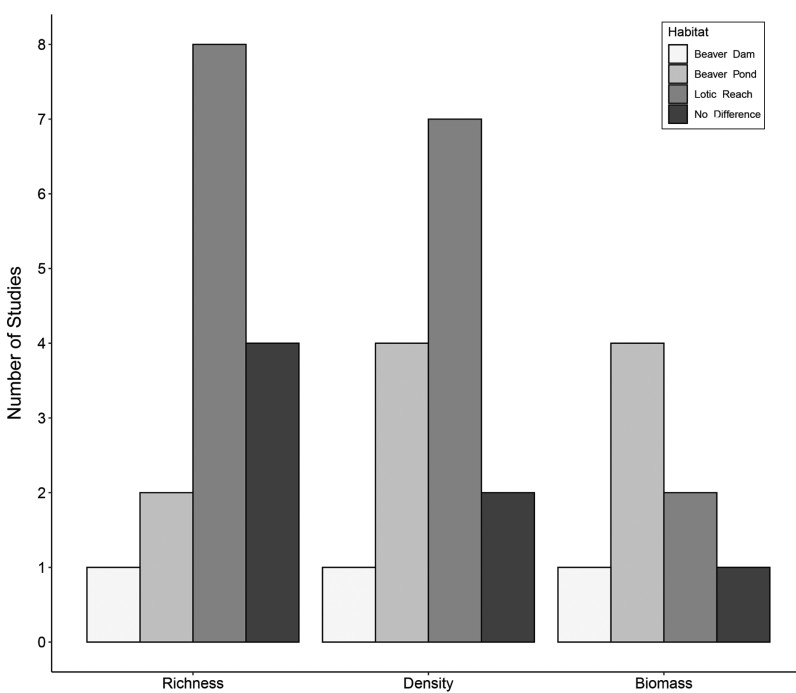

**Figure 2 Trends in species richness, density, and biomass of aquatic macroinvertebrates in beaver-altered streams.** Number of studies reporting highest aquatic macroinvertebrate richness, density and biomass per habitat.

(FFG) in lotic reaches or beaver ponds. We acknowledge that all the studies utilized different sampling methods, so we are only comparing their general trends in this scoping review to highlight future research avenues.

# PATTERNS AND COMMUNITY SHIFTS

## Richness and other biodiversity responses to beaver engineering

Fifteen studies compared aquatic macroinvertebrate richness between beaver-dammed lentic habitats and adjacent or nearby lotic habitats. Of those fifteen studies, eight (53%) concluded that lotic habitats had greater macroinvertebrate richness, whereas only two (13%) reported greater richness in beaver ponds and one (7%) in the beaver dam itself. Four studies (27%) found no difference between richness in lotic and beaver pond habitats (Fig. 2). These results suggest that unmodified lotic reaches tend to have higher species richness than dammed beaver ponds.

More broadly, many studies report increases in regional (gamma) aquatic macroinvertebrate biodiversity with the addition of beaver-created habitats (*Harthun, 1999*; *Kukuła et al., 2008*; *Arndt & Domdei, 2011*; *Law, Mclean & Willby, 2016*; *Czerniawski & Sługocki, 2018*; *Osipov, Bashinskiy & Podshivalina, 2018*; *Law et al., 2019*; *Bush et al., 2019*; *Robinson et al., 2020*; *Washko, Roper & Atwood, 2020*; *Wojton & Kukuła, 2021*) due to increased habitat heterogeneity (*e.g.* adding various lentic areas to lotic systems, increasing woody debris patches, constructing shallow canals) as well as within-patch heterogeneity (*e.g.* depths and inundation extent). Only one study found that the

beaver pond aquatic macroinvertebrates were a subset of the lotic community (*i.e.* not enhancing diversity), which was in Cape Horn, Chile, where beavers (*C. canadensis*) are non-native (*Anderson & Rosemond, 2007*). The beaver pond habitats created in Cape Horn were likely not distinct enough from the surrounding bog habitat to enhance species diversity in the same way as occurs in beavers' native ranges (*Anderson & Rosemond, 2007*).

Understanding how beaver engineering actions alter biodiversity is important to conservation and management of both the beavers themselves and to the species that are influenced by their ecosystem engineering. Having beaver ponds covering a variety of successional stages (*e.g.* abandoned *vs* active, or old *vs* new, and the continuum in-between) and ponds on a variety of stream sizes within a watershed (headwater/1st order streams as well as 2nd–4th order streams) can provide the foundation for a metacommunity effect, allowing macroinvertebrate recolonization after extreme events (*Wissinger & Gallagher, 1999*; *Hood & Larson, 2014*; *Nummi et al., 2021*). Further, *Naiman, McDowell & Farr (1984)* reported a disused beaver pond had higher macroinvertebrate diversity and biomass than the active ponds and riffles (1984), demonstrating that all successional stages of beaver habitats can uniquely contribute to regional aquatic macroinvertebrate diversity.

## Density & biomass in beaver-created habitats

The hydrogeomorphological changes associated with beaver engineering alter the habitat structure and food resources for macroinvertebrates. For example, shifts in the benthic substrate composition can shift the availability of surfaces for biofilm growth, and changes in water velocity can alter patterns of organic matter deposition (*Hodkinson, 1975b*). These types of changes will cause shifts in aquatic macroinvertebrate density and biomass. The extent of such shifts is unclear because although fine sediment deposition precludes colonization by many taxa (*Mackay, 1992*), the area of habitat and its vertical complexity may both increase (*McDowell & Naiman, 1986*; *Robinson et al., 2020*). Fourteen studies assessed aquatic macroinvertebrate density (either quantitatively or semi-quantitatively; the most commonly used sampling apparatus were a D-net and an Ekman grab) within beaver ponds and in adjacent lotic reaches. Of those studies, seven (50%) reported higher density in the lotic reaches, one study (7%) recorded the highest density within the beaver dam itself, and four studies (29%) found higher density within beaver ponds (Fig. 2). Two studies (14%) concluded beaver ponds and lotic reaches had the same aquatic macroinvertebrate density.

Fewer studies compared aquatic macroinvertebrate biomass between beaver ponds and adjacent lotic habitats (most commonly based on sampling using an Ekman grab). Of the eight studies, four (50%) found higher biomass in beaver ponds, one (12%) recorded higher biomass in the beaver dam itself, two (25%) reported higher biomass in the lotic reach, and one (12%) concluded there were no differences in aquatic macroinvertebrate biomass between beaver ponds and lotic reaches (Fig. 2).

These findings demonstrate the variability in how beaver engineering affects aquatic macroinvertebrate density and biomass. However, many studies posit that macroinvertebrate

density and biomass are related to how much organic matter is trapped in the pond or the lotic reach, with more organic matter leading to more macroinvertebrates (*McDowell & Naiman, 1986*; *Anderson & Rosemond, 2007*; *Arndt & Domdei, 2011*). Effects are likely highly context-dependent due to each stream's differing geological, topographical, ecological, and geographical setting which shape the geomorphological changes that follow dam building and the associated biological responses.

## Community composition and functional feeding groups (FFG) response to beaver activity

We expected that changes in habitat would prompt significant compositional shifts in the aquatic macroinvertebrate community. Of the 20 studies that examined differences in macroinvertebrate community composition between beaver ponds and lotic reaches, all reported the communities were indeed different. Due to broad geographic differences amongst the studies, the specific taxa cannot be directly compared, but their habitat affiliations can be viewed in Table S1. Generally, lentic-type species groups such as Odonata, Chironomidae, Dytiscidae, and Mollusca were more often associated with beaver pond habitats, while Elmidae and Plecoptera were, unsurprisingly, associated with lotic reaches. Only one study involved beaver pond successional stages within a stream system, reporting differences in the taxa present within beaver ponds of different ages (*Bush et al., 2019*).

Approximately half of the studies investigating aquatic macroinvertebrates in beaver-altered streams assessed differences in FFGs between beaver ponds and lotic reaches (Table S1). However, of these studies, not all reported results for each FFG and they reported results in different ways. Here, we synthesize the relative presence for each group within each habitat based on which groups each article emphasized as more important or more abundant.

Eleven studies included predatory FFGs, comprising engulfers, piercers, or both groups combined into one category of predators. Of these studies, eight (73%) documented an increase in predators within beaver ponds relative to lotic reaches (Fig. 3). Three studies (27%) showed no difference in predators between habitats. Increases in predators within beaver ponds may be due to easy-to-catch prey—specifically, drifting macroinvertebrates that become stranded in low-velocity beaver ponds with few interstitial spaces in which to hide (*Washko, Roper & Atwood, 2020*). Other explanations are that the beaver ponds produce large quantities of detritivorous taxa, which support numerous macroinvertebrate predators (*Harthun, 1999*), or that submerged vegetation within the ponds is suited to sit-and-wait predators such as Odonata (*Hann, 1995*; *Sychra, Adámek & Petřivalská, 2010*).

Nine studies accounted for shredders. Two studies (22%) showed more shredders within beaver ponds compared to lotic reaches (Fig. 3). Four (44%) documented higher numbers of shredders within lotic reaches relative to beaver ponds. Lastly, three (33%) studies found no differences in shredders between habitats. These disparate findings may reflect intrinsic differences between sites in the volume and quantity of allochthonous inputs from the riparian zone (*Cummins et al., 1989*). Specifically, if the riparian areas around beaver ponds are more open because beavers have removed woody vegetation, less

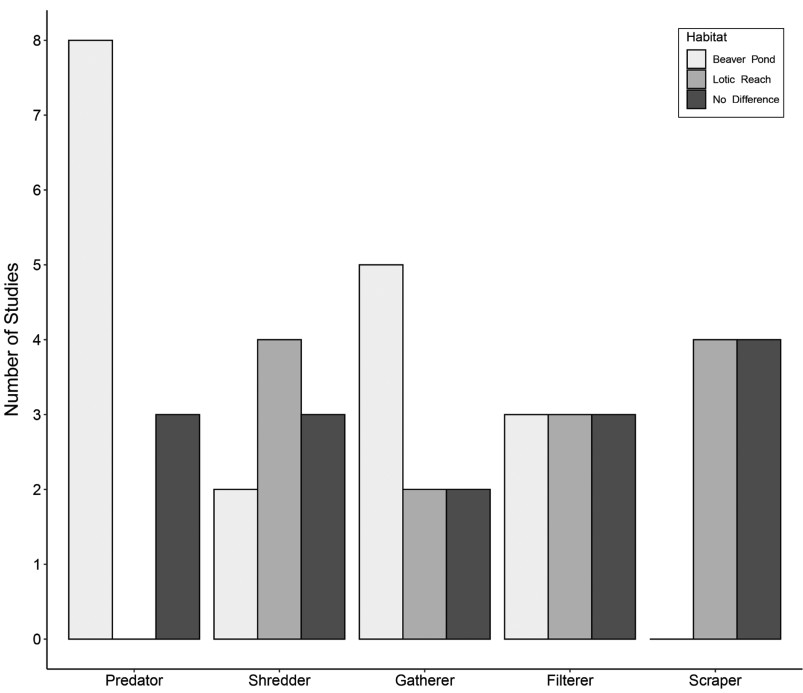

**Figure 3 Trends in functional feeding group abundances in beaver-altered streams.** Number of studies reporting higher abundance of each functional feeding group (FFG) in each habitat.

organic matter may enter ponds relative to the canopy-covered lotic reaches, making beaver ponds less conducive to shredders. However, organic matter from upstream often accumulates in the low-velocity beaver ponds (*Hodkinson, 1975a*, *1975c*; *McDowell & Naiman, 1986*; *Margolis, Raesly & Shumway, 2001*), which may support higher shredder densities. One could hypothesize that shredders may be abundant in new beaver ponds, where there is plenty of dead coarse organic matter in the form of dying plants (or ponds with a recent change in inundation level, *e.g.*, *Hood, McIntosh & Hvenegaard, 2021*). Alternatively, lotic reaches with large interstitial spaces may trap more organic matter (*Hoover et al., 2010*). Consequently, shredder patterns are likely context-dependent.

Nine articles assessed gatherer taxa. Of these, five (55%) saw more gatherers within beaver ponds compared to lotic reaches (Fig. 3). Two studies (22%) showed fewer gatherers in beaver ponds relative to lotic reaches. Lastly, two studies (22%) reported no differences in gatherers between habitats. Gatherer density within beaver ponds may generally be greater than the associated lotic reach because fine particulate organic matter (FPOM) is deposited in low-energy habitats like beaver ponds (*McDowell & Naiman, 1986*), providing food for gatherers (*Cummins & Klug, 1979*). However, the pattern is variable because gatherers can also be found in high densities downstream of dams where organic matter has leaked from the impoundment (*Smith et al., 1991*; *Redin & Sjöberg, 2013*).

Nine publications reported on filterer taxa. Of these, three studies (33%) documented more filterers within beaver ponds relative to lotic reaches (Fig. 3). Three (33%) demonstrated fewer filterers within beaver ponds compared to lotic reaches, with the

remainder recording no difference between habitats. This variability is surprising given that greater flow velocities in lotic reaches should generally transport more suspended food resources such as FPOM to filterers' nets or mouthparts (*Cummins & Klug, 1979*). However, one study suggested that the loss of velocity within beaver ponds causes the FPOM to fall out of suspension, supporting benthic-dwelling Mollusca (*Harthun, 1999*). Further, the complex woody structures of beaver dams that impound ponds are known to trap FPOM while water flows through (*Redin & Sjöberg, 2013*), thereby supporting filterer populations (*Clifford, Wiley & Casey, 1993*; *Rolauffs, Hering & Lohse, 2001*). Differences in the composition, age, or structural integrity of dams may also contribute to variable responses of filterers, though this was not mentioned in the assessed studies.

Finally, eight studies accounted for scraping taxa. Of these, four (50%) recorded lower scrapers within beaver ponds compared to lotic reaches (Fig. 3). The other four (50%) showed no differences in scrapers between beaver ponds and lotic reaches. Beaver ponds may have fewer scrapers because fine sediment replaces and covers coarser substrates (*Anderson & Rosemond, 2007*; *Washko, Roper & Atwood, 2020*), resulting in less surface area for biofilm growth. However, some beaver ponds could maintain scraper populations despite the sediment deposition due to the addition of beaver-associated woody debris (*e.g.* food caches, discarded branches, fallen deadwood from inundated trees), or aquatic plant substrates, providing the necessary scraping surface for macroinvertebrates (*Hering et al., 2001*; *Benke & Wallace, 2003*). The density of scrapers is likely contextually dependent on the sediment deposition processes and the beaver's placement of food caches and other woody debris.

These highly variable results demonstrate the importance of the local and regional context in shaping the FFG composition of beaver-created habitats. While predatory aquatic macroinvertebrates seem to be more prominent in beaver ponds relative to the lotic habitats, the other FFGs did not display consistent patterns. This is likely because of local factors shaping food resources such as availability of surfaces for biofilm growth, types and quantities of leaf litter, extent of residual tree shading, increases in DOM due to impoundment, changes in ability to suspend FPOM due to water velocity, or changes in composition and size structure of fish populations. Rarely are all these aspects fully quantified alongside macroinvertebrate studies.

Moreover, the effects of being located downstream of a beaver dam may also influence macroinvertebrate density. One study reported more predators and gatherers in the benthos downstream of the dam, positing that the exported matter from the impoundment enhanced macroinvertebrate densities, though an increase in precipitated metals prohibited high filterer colonization until further downstream (*Smith et al., 1991*). Another study also reported more filterers than gatherers in the drift downstream of the dam, arguing that the filterers must stay in the beaver dam itself to siphon suspended FPOM, while the gatherers colonize directly downstream to collect the exported FPOM (*Redin & Sjöberg, 2013*). The resuspension of organic matter due to beaver disturbances (moving woody debris to and within the pond, canal digging, or dam and lodge maintenance) could be expected to increase downstream FPOM, but further studies are needed to confirm this. Lastly, a small project on the inflows and outflows of beaver meadows also reported more

scrapers upstream and more filterers downstream, citing availability of food resources as the most probable underlying mechanism (*Doebley, 2020*).

Conversely, beaver dams may also partially block downstream drift of invertebrates, preventing some taxa from colonizing in high numbers below the dam. For example, one study documented more Ephemeropterans upstream of the dam, citing the pond as a drift-trapping mechanism preventing them from joining the drift downstream (*Redin & Sjöberg, 2013*). Conversely, a study comparing beaver pond aquatic invertebrate communities pre- and post- pond-leveler installation found few differences (*Hood, McIntosh & Hvenegaard, 2021*). Omnivore feeding groups decreased and shredders increased, which was attributed to the increase in shoreline vegetation habitat after the water levels were lowered. However, allowing more flow through the beaver dam, *via* a pond leveling device, did not change species composition or diversity (*Hood, McIntosh & Hvenegaard, 2021*).

In summary, streams that are beaver-altered have a greater habitat heterogeneity and therefore a greater gamma diversity and functional redundance, which should lend itself to increased resilience. Yet trends in alpha diversity and functional feeding responses vary amongst the habitats and the specific context in which they exist.

## RESEARCH GAPS

To better understand the interactions between beaver-altered habitats and aquatic macroinvertebrate communities, we have identified the following research gaps from the reviewed literature.

### Ecosystem processes

Aquatic macroinvertebrate communities are inextricably linked to ecosystem functions (*Wallace & Webster, 1996*), so changes in the macroinvertebrate community and its associated resource base due to beaver engineering will, by inference, alter ecosystem processes (*Anderson & Rosemond, 2007*). For example, shifts in the aquatic macroinvertebrate community can result in shifts in nutrient dynamics (*Atkinson et al., 2017*; *Balik et al., 2018*). However, few studies have investigated these questions within beaver ponds and the findings for FFGs presented here are based on only eight to ten studies per group and were cross-sectional in design. Further research is needed to assess changes in organic matter processing, primary productivity, and nutrient cycling within beaver ponds relative to lotic reaches over relevant time periods (*i.e.* several years to the lifespan of a beaver dam, potentially decades), and in contexts where the landscape matrix varies. Given the high spatial variation in water temperature regimes within a beaver-modified stream (*Majerova et al., 2015*), physiochemical properties may be similarly heterogeneous at fine scales.

Food web processes may also be affected both within beaver ponds and in terrestrial areas that receive aquatic subsidies or experience beaver foraging (*Milligan & Humphries, 2010*). If beaver-altered habitats change macroinvertebrate composition or densities, predator-prey dynamics may change for fish (*Kemp et al., 2012*), waterfowl (*Nummi, 1992*), and other consumers reliant on subsidies from aquatic to terrestrial habitats such as mice, shrews, bats, and riparian spiders or carabid beetles (*Hering & Plachter, 1997*;

*Nummi et al., 2011*; *McCaffery & Eby, 2016*; *Sundell, Liao & Nummi, 2021*). Very few macroinvertebrate studies look beyond the beaver pond; only five studies we reviewed quantified aquatic insect emergence. These types of data are valuable for conservation purposes in terrestrial habitats receiving subsidies of aquatic insects (*Bartrons et al., 2013*). For example, data on how endangered or threatened riparian species are affected by changes in insect emergence within beaver impoundments can aid in conservation efforts, such as for the Southwestern Willow Flycatcher (*Empidonax traillii extimus*; *Finch & Stoleson, 2000*) and European pond bat (*Myotis dasycneme*; *Nummi et al., 2011*). More studies on beaver-altered riparian zones and their consequences for lateral connectivity would elucidate nuances in food web changes. Moreover, embracing novel technologies such as eDNA metabarcoding (*Harper et al., 2019*) will make these research projects and monitoring efforts more feasible and less taxonomically biased.

## Aquatic invertebrate quantification

A total of 61% of studies included aquatic macroinvertebrate density, and only 35% included aquatic macroinvertebrate biomass. Furthermore, of the 23 studies reviewed, 14 used quantitative macroinvertebrate sampling methods (*e.g.* Eckman or core sample; four of which were solely emergence measurements using emergence traps), eight used semi-quantitative methods (*e.g.* D-net sweeps), and one paper used quantitative sampling in lotic habitats and semi-quantitative in lentic habitats. Although sampling quantitatively in ponds can be difficult due to varying depths, substrate or vegetation characteristics, and lack of directional flow, doing so is important for documenting shifts in density. Species quantity is an undervalued aspect of biodiversity relative to richness and will complement insights into beaver-induced habitat changes. Therefore, quantitatively documenting aquatic macroinvertebrate biomass and density shifts will provide better support for ecosystem function and food web studies.

## Freshwater biodiversity crisis

While it has been established that beavers can enhance habitat heterogeneity and macroinvertebrate diversity (*Willby et al., 2018*; *Law et al., 2019*), we have also established that changes to aquatic macroinvertebrate community composition are highly context-dependent. In the face of the freshwater biodiversity crisis (*Albert et al., 2020*), scientists need a better understanding of the underpinning effects of beavers on biodiversity in different countries, stream types, geologies, landscapes etc, and, importantly, at different positions on the human impact gradient. Beaver ponds can be very different from non-beaver ponds in terms of habitat structure and the species they support (*Bush & Wissinger, 2016*; *Willby et al., 2018*; *Nummi et al., 2021*). Also, other adjacent beaver-created habitats contribute to beaver-associated habitat heterogeneity at different scales. Beaver canals, for example, provide habitat for macroinvertebrate predator species otherwise absent from the waterbody. These canals can support high macroinvertebrate biodiversity, and also aid in amphibian dispersal (*Grudzinski, Cummins & Vang, 2020*).
Further, individual beaver dams vary greatly in structure and hydrologic context, affecting ecosystem resilience in different ways (*Ronnquist & Westbrook, 2021*). The wood of the dams themselves or other beaver-associated woody debris (*e.g.* felled trunks or fallen deadwood) creates highly-structured lentic zones within streams, sometimes greatly amplifying the faunal effects of a natural debris pile, or introduces microhabitats that differ physically or in their resource value and which therefore suit different taxa (*Hering et al., 2001*). The grazing and trampling activities of a relatively large herbivore are also an important element of habitat heterogeneity in their own right, independent of more conventional engineering (*Willby et al., 2018*). Having numerous examples of how beavers (and humans simulating the effects of beavers, such as with beaver dam analogues) change stream macroinvertebrate communities under a range of conditions will both improve scientific knowledge and aid support for the role of ecosystem engineers more generally in mitigating the freshwater biodiversity crisis.

One particular condition for further study is stream gradient. Fourteen studies mention the general stream gradient of their study sites, but only one directly studied beaver-altered streams of contrasting gradients (*Robinson et al., 2020*). Stream gradient is important to consider because of its profound effects on beaver pond morphology. High-gradient streams may show less physical change post-damming because flow remains higher and ponds are inevitably relatively small (although dam densities may be high), while low-gradient streams may be turned into large wetland complexes (*Robinson et al., 2020*). The degree of habitat change translates to changes to aquatic macroinvertebrates. For example, if high-gradient areas undergo less morphological change after beaver reintroduction, the effects and benefits for regional biodiversity may be reduced relative to those seen in low-gradient systems. Further investigation of elevation profiles and gradients in beaver complexes may be of interest if land managers need to prioritize support for a specific taxa or management objective.

## Downstream effects

Due to the complex manner in which beaver dams affect lotic habitats directly downstream, more research may elucidate how changes in flow, temperature or dissolved oxygen regimes, or organic matter availability can alter aquatic macroinvertebrate community composition. First, beaver-induced hydrologic changes can affect macroinvertebrates in downstream undammed, lotic segments in addition to the ponded areas. For example, aquatic macroinvertebrate colonization has been affected by altered stream discharge patterns (*Schlosser, 1995*). Further, beaver dams can change groundwater hydrology, resulting in colder water temperatures downstream of dams that stimulate mayfly growth and fecundity (*Fuller & Peckarsky, 2011*). Lastly, the reaches directly downstream of dams can have higher biodiversity (*Wojton & Kukuła, 2021*), and, as mentioned previously, different FFGs and aquatic macroinvertebrate densities can be found above and below beaver dams due to changing food resources (*Smith et al., 1991*; *Redin & Sjöberg, 2013*). As the finer mechanics of beaver-altered hydrology,

geomorphology, and biogeochemistry become better resolved (*e.g. Brazier et al., 2021*; *Larsen, Larsen & Lane, 2021*), scientists can apply this understanding specifically to predict effects on aquatic macroinvertebrates.

### Invertebrates of beaver-altered lakes and wetlands

Compared to studies in beaver-altered streams and rivers, there are few studies investigating the aquatic macroinvertebrate communities of beaver-altered wetlands and lakes (*Hood & Larson, 2014*; *Bush & Wissinger, 2016*; *Willby et al., 2018*; *Law et al., 2019*; *Bashinskiy, 2020*). These studies indicate that wetland and lake macroinvertebrates increase in diversity and experience community shifts due to increased habitat heterogeneity through woody debris inputs and canal building (*Hood & Larson, 2014*; *Bashinskiy, 2020*), much like riverine macroinvertebrates. Further, beaver-occupied wetland taxa differ from those of other nearby wetlands (*Willby et al., 2018*; *Law et al., 2019*; *Nummi et al., 2021*). Lastly, beavers can change the successional trajectory and hydroperiod of wetlands, altering community composition, metacommunity dynamics (*Wissinger & Gallagher, 1999*; *Hood & Larson, 2014*; *Nummi et al., 2021*), and lateral connectivity. These studies demonstrate that the beaver's role as a restoration agent is also applicable in wetland contexts and deserves further investigation. For example, space-for-time or long-term studies of succession in beaver dam complexes or meadows would demonstrate how or if specific benefits or effects persist, and for how long.

## CONCLUSIONS

Beaver engineering affects aquatic macroinvertebrate communities in rivers. Lotic reaches often have higher species richness compared to ponded areas, but overall stream biodiversity increases with the addition of ponded habitats. Similarly, lotic reaches may have higher aquatic macroinvertebrate density while beaver ponds have higher biomass, and beaver ponds often contain more predatory aquatic macroinvertebrate species than lotic reaches. However, given that all beaver damming reduces stream energy and creates depositional environments, the biological changes reported were less predictable than might be expected. As beavers continue to recolonize their former ranges, researchers will undoubtedly reveal more about their effects on aquatic macroinvertebrates and cascading effects on ecosystem functioning, providing a glimpse into the former natural state of landscapes and their potential for recovery.

## ACKNOWLEDGEMENTS

Our manuscript was improved based on comments from Petri Nummi and one other anonymous reviewer. We thank these reviewers for their contributions.

### Funding

The authors received no funding for this work.

## Competing Interests

The authors declare that they have no competing interests.

## Author Contributions

- Susan Washko conceived and designed the experiments, performed the experiments, analyzed the data, prepared figures and/or tables, authored or reviewed drafts of the paper, and approved the final draft.
- Nigel Willby conceived and designed the experiments, authored or reviewed drafts of the paper, and approved the final draft.
- Alan Law conceived and designed the experiments, prepared figures and/or tables, authored or reviewed drafts of the paper, and approved the final draft.

## Data Availability

The data are available in the Supplemental Files.

## Supplemental Information

Supplemental information for this article can be found online at http://dx.doi.org/10.7717/peerj.13180#supplemental-information.

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
