# Peer review of "How beavers affect riverine aquatic macroinvertebrates: a review"

_PeerJ, doi:10.7717/peerj.13180_

## Round 0.1 · original submission · Major Revisions

Dear Dr. Washko and colleagues:

Thanks for submitting your manuscript to PeerJ. I have now received two independent reviews of your work, and as you will see, the reviewers raised some concerns about the research. Despite this, this the reviewers are optimistic about your work and the potential impact it will have on research studying stream ecology and the impact of beavers on macroinvertebrate diversity. Thus, I encourage you to revise your manuscript, accordingly, taking into account all of the concerns raised by both reviewers.

Please provide more clarity where it is lacking per the reviewers’ suggestions. This is especially needed in describing the overall methodological approach.

There are many minor suggestions to improve the manuscript (typos, nuances, etc.).

I agree with many of the concerns of the reviewers, and thus feel that their suggestions should be adequately addressed before moving forward.

I look forward to seeing your revision, and thanks again for submitting your work to PeerJ.

Good luck with your revision,

-joe

·

Basic reporting

Very neat and pleasant introduction, giving good background in a nutshell. The theme is interesting, and has not been recently reviewed.

Experimental design

The study questions are clear and relevant, and the search process well enough described. Limiting the review to lotic habitats on one hand leaves out some interesting aspects, on the other makes the review clear and concise. Moreover, the authors wisely still discuss some aspects, which have not been studied in lotic setting but are relevant, on the basis of studies made in lentic wetlands.

Validity of the findings

The review well summaries the existing knowledge of the effects of beavers on invertebrates on lotic habitats. And, it identifies a gap in that the lentic habitats are much less studies in this respect. Connected to this, it would be interesting to see a pondering about what s the share of the beaver ponds made in lentic and lotic habitats. I know some studies in limited areas of this kind, but I wonder if there would be a possibility to make a crude large scale estimate of this matter?

Additional comments

Individual points:

Patterns and Community Shifts

Richness and other biodiversity
Here you could consider mentioning an experimental study from Finland (Nummi 1989) which shows the shift in invertebrate community in a creek section upon a damming imitating beaver, both in the littoral and creek bed as well as showing the change in number of emerging insects. It also demonstrates the substantial flow of organic terrestrial matter to the water in the from tree leaves in the early phase of inundation.
Maybe here you could mention also the considerable within patch heterogeneity, including deep water behind the dam, flooded swamps, shallow marshes etc., noted by e.g. Bush & Wissinger 2016 and Willby et al. 2018. This also related to the discussion later, in lines 356-363, about “low-gradient streams may be turned into large wetland complexes”.
l. 150-151. Missing a word? These types of changes will cause shifts IN aquatic macroinvertebrate density and biomass.

Feeding groups
In the discussion of the abundance of different feeding groups it could be relevant to shortly deal with effect succession on these; e.g. shredders may be abundant in new beaver ponds where there is plenty of dead coarse organic matter in the form of dying plants, or predatory inverts might be abundant in new flowages because of little competition/predation from fishes.

Research Gaps
Ecosystem processes
l. 313. Along with birds, maybe bats as a group, or e.g. near threatened species Myotis dasycneme could be mentioned here.

Aquatic insect quatification
Here it was interesting to note the high number of predators in spite of the fact that quite much e.g. corers were used which hardly trap many of the very mobile predatory insects. I m not sure if this needs any comment.

Freshwater biodiversity crisis
l. 341. Maybe Bush &Wissinger 2016 could be added as reference here.
l. 347-349. Here, in the connection of the mere structural things, you might consider mentioning dead trees, especially trunks fallen to the water.

Reviewer 2 ·

Basic reporting

1/ -The authors conduct an interesting literature review on how beaver activity affects aquatic macroinvertebrates. I applaud the idea and its broad scale, but have reservations about the manuscript. However, I think that refinement of the article is required to improve clarity, and I provided some suggestions to achieve this.

Experimental design

2/ -I commend the authors for the level of detail with some of their literature reviews, however some important references are missing.

Validity of the findings

3/ -The authors should revise/clarify the research questions.
4/ -More explanations and interpretations must be added for the results, which are not enough. To sum up - I think this could be a very interesting review, but is hard to follow the argumentation as it stands.

Additional comments

Title.
5/ -Instead of "A synthesis of global data", just "A review" (it is more adequate).

Abstract.
6/ -Headings in a structured abstract would increase its clarity. Please remember, that headings in structured abstracts (i.e., Background. Methodology. Results. Conclusions) should be bold and followed by a period.

Introduction.
7/ -In general, the Introduction doesn’t really summarise the whole manuscript. I would recommend that the authors re-design to succinctly describe in more detail the work that they set out to do. Some important references are missing.
8/ -Line 46-47. ‘...becoming increasingly interested in how beaver engineering activities alter stream ecosystems within a modern landscape context’ - see Brazier et al., 2021
9/ -Line 48. ‘…can improve salmonid fish habitat’ - see also Kukuła & Bylak, 2011
10/ -Line 69. ‘…can support different aquatic macroinvertebrate taxa’ – see also Bylak et al., 2021 - support for endangered mussels.

Research aims.
11/ -Line 81-85 - Point 1 and 2: These are the objectives of the research paper, not the review paper.
12/ -My recommendation is to, (A), revise/clarify the research questions. For example the second research question presented is: “Determine if the aquatic macroinvertebrate communities differ in composition between beaver ponds and lotic stream segments” I could not see how this research questions was addressed by the analyses conducted. Three research questions stand out (based on the data collected): (1) What is the state of knowledge in understanding of taxonomic and functional changes in benthic macroinvertebrate assemblages of the beaver-altered streams? How has the field of knowledge evolved over time? (2) Which types of associations between beaver activity and aquatic invertebrate taxa have been described to date? What are the most relevant trends or patterns? (3) Knowledge gaps (which are good as they are). Please modify also the Abstract section accordingly.

RICHNESS AND OTHER ...
13/ -Line 139 – ‘a metacommunity effect’ – metacommunities in the beaver dam-and-pond complexes have been previously described in detail in fish populations - see Bylak & Kukuła, 2018.

INVERTEBRATES OF BEAVER-ALTERED LAKES AND WETLANDS
14/ -Line 383-397 - Is this whole paragraph really needed ??? (see - Title and Survey methodology)

References.
15/ Following, you will find some new related references which should be added to literature set:
-Brazier et al. 2021. Beaver: Nature's ecosystem engineers. doi: 10.1002/wat2.1494
-Bylak et al., 2021. Potential use of beaver Castor fiber L., 1758 dams by the thick-shelled river mussel Unio crassus Philipsson, 1788. doi:10.1080/13235818.2019.1664371
-Bylak & Kukuła, 2018. Living with an engineer: Fish metacommunities in dynamic patchy environments. doi:10.1071/MF17255
-Kukuła & Bylak, 2011. Ichthyofauna of a mountain stream dammed by beaver. doi: 10.2478/v10086-010-0004-1

Figures.
16/ -A flowchart should be added to the article to show the research methodology.
17/ -Please also increase the font size of numbers in Figure 1.

---

## Round 0.2 · accepted · Accept

Dear Dr. Washko and colleagues:

Thanks for revising your manuscript based on the concerns raised by the reviewer. I now believe that your manuscript is suitable for publication. Congratulations! I look forward to seeing this work in print, and I anticipate it being an important resource for groups studying stream ecology and the impact of beavers on macroinvertebrate diversity. Thanks again for choosing PeerJ to publish such important work.

Best,

-joe

·

Basic reporting

ok

Experimental design

ok

Validity of the findings

ok

Additional comments

The authors have well replied to the points I raised.

Reviewer 2 ·

Basic reporting

The paper has been improved, focusing on the key messages of the paper.

Experimental design

The title and abstract are much better. Paper objectives are better, and integrating strong perspectives following the results obtained.

Validity of the findings

Paper objectives are better, and integrating strong perspectives following the results obtained.

Additional comments

Most of the suggestions made have been taken on board by the authors. I support its publication in its current form.